# Nephrocalcinosis: A Review of Monogenic Causes and Insights They Provide into This Heterogeneous Condition

**DOI:** 10.3390/ijms21010369

**Published:** 2020-01-06

**Authors:** Fay J. Dickson, John A. Sayer

**Affiliations:** 1Hull University Teaching Hospitals, Anlaby Road, Hull HU3 2JZ, UK; fayhill@gmail.com; 2Translational and Clinical Research Institute, Faculty of Medical Sciences, Newcastle University, Central Parkway, Newcastle upon Tyne NE1 3BZ, UK; 3The Newcastle upon Tyne NHS Hospitals Foundation Trust, Newcastle upon Tyne NE7 7DN, UK; 4NIHR Newcastle Biomedical Research Centre, Newcastle upon Tyne NE4 5PL, UK

**Keywords:** nephrocalcinosis, nephrolithiasis, hypercalciuria, monogenic, precision medicine

## Abstract

The abnormal deposition of calcium within renal parenchyma, termed nephrocalcinosis, frequently occurs as a result of impaired renal calcium handling. It is closely associated with renal stone formation (nephrolithiasis) as elevated urinary calcium levels (hypercalciuria) are a key common pathological feature underlying these clinical presentations. Although monogenic causes of nephrocalcinosis and nephrolithiasis are rare, they account for a significant disease burden with many patients developing chronic or end-stage renal disease. Identifying underlying genetic mutations in hereditary cases of nephrocalcinosis has provided valuable insights into renal tubulopathies that include hypercalciuria within their varied phenotypes. Genotypes affecting other enzyme pathways, including vitamin D metabolism and hepatic glyoxylate metabolism, are also associated with nephrocalcinosis. As the availability of genetic testing becomes widespread, we cannot be imprecise in our approach to nephrocalcinosis. Monogenic causes of nephrocalcinosis account for a broad range of phenotypes. In cases such as Dent disease, supportive therapies are limited, and early renal replacement therapies are necessitated. In cases such as renal tubular acidosis, a good renal prognosis can be expected providing effective treatment is implemented. It is imperative we adopt a precision-medicine approach to ensure patients and their families receive prompt diagnosis, effective, tailored treatment and accurate prognostic information.

## 1. Introduction

Nephrocalcinosis can broadly be defined as the deposition of calcium, either as calcium phosphate or calcium oxalate, within the interstitium of the kidney [1]. Whilst Oliver Wrong’s original classification sub-divided nephrocalcinosis into being either molecular, microscopic or macroscopic, in clinical practice the term commonly refers to macroscopic nephrocalcinosis that can be detected radiologically [2]. Rarely, asymmetric presentation of nephrocalcinosis localised to the renal cortex may represent calcium release from tissue breakdown, for example in renal transplant rejection or renal infarction [3]. However, in the majority of cases, nephrocalcinosis affects the renal medulla and can be detected as bilateral, symmetrical increased echogenicity within the renal pyramids on ultrasound imaging [4]. This non-invasive imaging modality is utilised either as a screening tool, or as a method of assessing disease progression/response to treatment, as medullary nephrocalcinosis can be reliably graded (Grade I–III) according to the extent of increased echogenicity affecting the medullary pyramids [5]. 

Whilst the exact pathogenesis of nephrocalcinosis remains under investigation, it is acknowledged medullary nephrocalcinosis is a consequence of hypercalciuria. Increased urinary calcium load arises either through increased calcium absorption (extra-renal causes) or impaired calcium reabsorption within the renal tubule [2]. The majority of calcium reabsorption (~65%) occurs in the proximal tubule, whilst ~25% is reabsorbed in the thick ascending limb of the loop of Henle [6] and ~5% is reabsorbed from the cortical collecting duct [2]. Identification of monogenic causes of nephrocalcinosis affecting these areas has provided valuable insights into the pathogenesis of this heterogeneous condition. Interestingly, although a further ~7–10% of calcium is reabsorbed within the distal convoluted tubule, no monogenic causes of nephrocalcinosis have been identified which affect this section of the renal tubule [2].

Hypercalciuria, in addition to being part of the underlying pathological process for nephrocalcinosis, also predisposes patients to renal stone formation (nephrolithiasis). Nephrocalcinosis and nephrolithiasis will therefore commonly co-exist within the phenotypes of these rare monogenic conditions, and nephrolithiasis onset at a young age may prompt investigations for nephrocalcinosis and uncover an inherited condition [7].

Nephrocalcinosis in itself is a rare disorder, and consequently monogenic forms affect even smaller numbers of the population. For those individuals affected by nephrocalcinosis, however, a tailored, individualised approach to their initial diagnostic work-up is imperative. Presentation at a young age, a family history of an affected individual or carrier, or a history of consanguinity in the family is often suggestive of an inherited tubulopathy. Several factors, including recessive inheritance patterns and varied phenotype presentation, dictate that a family history is often not present, and suspicion of a monogenic cause should not be dismissed simply due to absence of the above factors.

The morbidity associated with nephrolithiasis and nephrocalcinosis is widely accepted [8]. Whilst an overall approach may focus upon slowing progression of associated chronic kidney disease (CKD), specific treatment strategies, such as dietary modifications or use of thiazide diuretics, will vary depending upon the underlying disease process. Widespread adoption of precision medicine within this field is likely to have significant advantages for both current and future patients. Several monogenic causes of nephrocalcinosis are associated with rapid progression of CKD, often requiring initiation of renal replacement therapies before adulthood. Early diagnosis of a monogenic cause is vital for providing accurate prognostic information for the patient and their family, including the opportunity to screen other family members or offer pre-implantation genetic diagnosis (PGD) testing for subsequent pregnancies. Perhaps most significantly, a prompt diagnosis may also avoid the individual being exposed to unnecessary or harmful treatments. For example, a patient with *OCRL* mutations (Dent disease 2) was reported to receive immunosuppressive therapies including corticosteroids and cyclophosphamide, based on a misdiagnosis of nephrotic syndrome, before their inherited tubulopathy was correctly identified [9]. Commonly used medications may easily inadvertently worsen the condition of patients with other inherited tubulopathies: loop diuretic use can potentiate hypercalciuria and worsen nephrolithiasis/nephrocalcinosis burden, whereas use of potassium-sparing diuretics should be avoided in patients with distal renal tubular acidosis [10]. Furthermore, in cases where a proactive approach to personalised genetic medicine has been taken, next generation sequencing has identified *CLCN5* mutations (Dent disease 1) in patients for whom only low-molecular weight proteinuria was present at diagnosis, i.e., before the full phenotype had emerged [11]. Finally, given the current paucity of treatment options for nephrocalcinosis, it is potentially from the study of these rare monogenic causes that key future therapeutic targets may emerge which could revolutionise our management of the condition (Table A1).

## 2. Monogenic Causes of Nephrocalcinosis

### 2.1. CLCN5 Mutations

The proximal tubule represents the site of greatest calcium reabsorption within the renal tubule, and it is mutations in the ClC-5 chloride transporter in this region, encoded by *CLCN5*, that give rise to Dent disease type 1 [12]. This condition demonstrates an X-linked recessive pattern of inheritance; affected males usually display a triad of low molecular weight (LMW) proteinuria, hypercalciuria and nephrocalcinosis, although the triad may be incomplete at initial presentation [11]. Female carriers are usually asymptomatic, but may occasionally have LMW proteinuria, hypercalciuria or nephrolithiasis [12].

Dent disease 1 (*CLCN5* mutations) accounts for 60% of Dent disease cases. *OCRL* mutations, which may result in a spectrum of phenotypes ranging from Dent disease 2 to the more severe Lowe oculocerebrorenal syndrome, account for a further 15% of cases, whilst the underlying genetic mutation remains unascertained in 25% of cases [12]. The varied phenotypes of Dent disease may also include a partial Fanconi syndrome as the initial clue indicating proximal tubular dysfunction [13], whilst nephrolithiasis or haematuria can also feature alongside the cardinal LMW proteinuria [11]. Although genetic conditions with a Fanconi syndrome, including Lowe oculocerebrorenal syndrome (*OCRL* mutations) and cystinosis (*CTNS* mutations) may also feature nephrocalcinosis [14,15], it remains more commonly associated with Dent disease 1 (*CLCN5* mutations) [16].

Patients with Dent disease carry a poor prognosis in terms of renal function: 30–80% of males develop end-stage renal disease (ESRD) by middle-age [12]. The development of CKD has been hypothesised to be linked to nephrocalcinosis, although this theory does not explain Dent disease patients without nephrocalcinosis who also develop progressive CKD, and it is probable more than one mechanism exists [16,17].

Treatment options for Dent disease are limited. Early diagnosis is crucial in order to prioritise preservation of renal function for as long as possible and offer accurate prognostic information [11]. Patients should undergo careful CKD monitoring with control of variables such as blood pressure. A strategy to try and reduce nephrocalcinosis development is to reduce urinary calcium excretion; thiazide diuretics may be used but in some patients their role is limited by unacceptable side effects including hypokalaemia, muscle cramps and dehydration [12]. The majority of patients will reach ESRD by the age of 40 [18]; renal transplantation offers the best outcomes as there is no recurrence in the renal allograft due to the donor kidney not carrying the causative mutation [12].

### 2.2. CYP24A1 Mutations

Nephrocalcinosis occurs as a result of impaired renal calcium handling; disorders of vitamin D metabolism can result in elevated calcium levels and represent a different pathway predisposing to nephrocalcinosis formation. The second stage of vitamin D activation takes place within the kidney, resulting in production of the active metabolite 1,25-dihydroxyvitamin D^3^. Mutations in *CYP24A1*, which encodes 1,25-hydroxyvitamin-D_3_-24-hydroxylase, result in an inability to catabolise this active vitamin D metabolite [19]. *CYP24A1* mutations were first detected following reports of a small cohort of babies developing adverse effects (including hypercalcemia and nephrocalcinosis) as a result of the public health intervention to routinely supplement formula milk with vitamin D [20]. Following further analysis, two distinct phenotypes resulting from *CYP24A1* mutations have been recognised, both of which frequently include nephrocalcinosis [19].

The first phenotype, idiopathic infantile hypercalcaemia (IIH) presents in childhood, and is classified as hypercalcaemia, infantile 1 (HCINF1). Affected infants often present with symptomatic hypercalcaemia, severe dehydration, vomiting and failure to thrive [21]. Nephrocalcinosis is often detectable on ultrasound imaging at diagnosis [21]. Management of these patients focusses upon removing exogenous sources of vitamin D (e.g., supplements), fluid resuscitation and future conservative measures (high fluid intake, dietary adjustments, avoidance of tanning beds) to prevent further nephrolithiasis/nephrocalcinosis formation.

A second, later-onset, phenotype has also been demonstrated in adults who present with nephrolithiasis, hypercalciuria or incidentally detected nephrocalcinosis [22]. This phenotype is usually less severe, and a history of vitamin D supplementation is not universally present. Treatment strategies focus upon low calcium and oxalate diet, avoidance of vitamin D supplementation and excessive sunlight exposure [19]. Use of azole antifungal agents (fluconazole, ketoconazole) have shown benefit as non-specific P450 enzyme inhibitors which inhibit 1α-hydroxylase (encoded by *CYP27B1*), thereby reducing production of the active form of vitamin D [19,23]. However, lifelong treatment with these agents may be undesirable owing to their side effect profile, including hepatotoxicity. Recently, rifampicin was demonstrated to reduce hypercalcaemia and effectively control 1,25-dihydroxyvitamin D^3^ levels in patients with *CYP24A1* mutations [24]. Rifampicin acts upon an alternative vitamin D catabolism pathway as a potent inducer of *CYP3A4* [24]. This provides an alternative therapy that may be better tolerated as a lifelong treatment.

### 2.3. SLC34A1 Mutations

Autosomal recessive inheritance of mutations in the sodium-phosphate co-transporter NaPi2a, encoded by *SLC34A1*, cause idiopathic infantile hypercalcaemia (IIH), classified as hypercalcaemia, infantile 2 (HCINF2), in a subgroup of patients without *CYP24A1* mutations [25]. In addition to the classic biochemical features associated with IIH of hypercalcaemia, hypercalciuria and high levels of 1,25-dihydroxyvitamin D_3_, patients with *SLC34A1* loss-of-function mutations exhibit hypophosphatemia [26]. This hypophosphatemia arises as a result of renal phosphate wasting within the proximal tubule, driving excessive production of 1,25-dihydroxyvitamin D_3_ and subsequent hypercalcaemia and hypercalciuria.

Nephrocalcinosis is a common phenotypical feature in patients with *SLC34A1* mutations [27]. In the acute presentation, where patients may have classical features of IIH such as failure to thrive, dehydration and vomiting, they should be fluid resuscitated and may receive loop diuretics (furosemide) to induce calciuresis [27]. Similar to patients with *CYP24A1* mutations, an azole antifungal agent (ketoconazole or fluconazole) may be used to inhibit 1α-hydroxylase, thereby reducing the high calcium levels predisposing to nephrocalcinosis through reduction in levels of 1,25-dihydroxyvitamin D_3_ [26]. The long-term management of these patients, however, should focus upon a low calcium diet and avoidance of vitamin D supplementation or heightened vitamin D exposure, in order to minimise the risk of nephrocalcinosis.

### 2.4. CLDN16 and CLDN19 Mutations

The thick ascending limb (TAL) of loop of Henle, where ~25% of calcium reabsorption [6] and ~60% of magnesium reabsorption usually takes place [28], is the site of two rare autosomal recessive channelopathies affecting calcium and magnesium absorption. Mutations in *CLDN16* and *CLDN19*, which encode the tight junction proteins claudin-16 and claudin-19 respectively, give rise to the condition familial hypomagnesaemia with hypercalciuria and nephrocalcinosis (FHHNC) [29].

Patients are symptomatic from a young age, although initial presenting features may be non-specific such as polyuria/polydipsia, failure to thrive and vomiting. The biochemical profile of FHHNC phenotypes includes excessive loss of calcium and magnesium in the urine, with normal serum calcium levels and low serum magnesium levels [28]. Importantly, serum hypomagnesaemia is not universally present, and a fractional excretion value of magnesium (FeMg%) should be calculated using serum magnesium, serum creatinine and urinary magnesium levels to ensure diagnostic accuracy [30]. Nephrocalcinosis is a universal feature which develops early in the disease course of FHHNC, and hypercalciuria is one predisposing factor for this [30]. A further predisposing factor for nephrolithiasis and nephrocalcinosis in FHHNC is hypocitraturia, which removes the protective effect of urinary citrate against precipitation of calcium salts in the urine [30]. The renal prognosis for FHHNC is poor; many patients progress to ESRD during adolescence [28]. Rapid progression of CKD in these patients may not be attributable solely to nephrocalcinosis, as their renal dysfunction is more severe/presents earlier than that observed in other tubulopathies predisposing to nephrocalcinosis such as primary distal renal tubular acidosis or Bartter syndrome [30]. *CLDN19* mutations are associated with severe ocular abnormalities as claudin-19 is also expressed in the retinal epithelium [28].

Treatment strategies for FHHNC patients focus initially on supportive measures aimed at reducing hypercalciuria and replacing magnesium; thiazide diuretics and oral magnesium supplements are used for this respectively, although their impact on total urine calcium and serum magnesium levels is not always significant [28,30]. Overall, these supportive treatments do not negate the progression of renal dysfunction, and renal replacement therapies are commonly necessitated before adulthood. Renal transplantation is the ideal option: the loss-of-function channelopathy is not present in the renal allograft so renal calcium and magnesium handling are normalised and there is no disease recurrence [31].

### 2.5. Bartter Syndromes

The Bartter syndromes describe five channelopathies affecting thick ascending limb (TAL) transporter proteins involved in sodium chloride (NaCl) re-absorption. Autosomal recessive inheritance of these gene mutations results in salt-losing tubulopathies characterised by excessive urinary sodium losses, with corresponding hypokalaemia, metabolic alkalosis and secondary hyperaldosteronism [32]. Nephrocalcinosis has been described in all Bartter syndromes but is most frequently associated with Bartter I, II and V [2]. The pathogenesis of nephrocalcinosis in Bartter syndromes is not fully understood, although it is most probably a consequence of hypercalciuria seen within the Bartter syndrome phenotypes [33]. Early identification of the Bartter syndrome genotype is clinically advantageous, especially given the fact nephrocalcinosis and other renal manifestations are not uniformly represented across the different Bartter syndrome phenotypes. Several clinical features, including age of onset, presence of transient hyperkalaemia, severity of hypokalaemia, sensorineural deafness and renal impairment may provide vital clues about the likely underlying genotype which can guide genetic testing.

Bartter syndromes I and II, caused by mutations in *SLC12A1* and *KCNJ1* respectively, usually present during the antenatal/postnatal period, and have been traditionally referred to as antenatal Bartter syndromes (aBS). Polyhydramnios, premature birth and low-birth weight are classic features of aBS, and nephrocalcinosis is frequently already detectable at this young age [34]. *KCNJ1* mutations, encoding the ROMK potassium channel, often display transient hyperkalaemia in the neonatal period prior to development of classic hypokalaemia [32]. Early treatment to correct electrolyte disturbance, rehydrate patients and minimise growth retardation is necessitated [35]. Cyclo-oxygenase inhibitors (e.g., indomethacin), which target the elevated prostaglandin levels seen within aBS, are an effective treatment and can lead to effective catch-up growth [33,35].

Our understanding of Bartter syndrome II has recently expanded to include a late-onset phenotype following case reports of two adults found to have *KCNJ1* mutations after incidental nephrocalcinosis detection. Both patients had nephrocalcinosis and mild renal impairment at diagnosis, and were treated with oral potassium supplementation, and either a potassium-sparing diuretic or angiotensin-converting enzyme inhibitor respectively [33,35].

Bartter syndrome III occurs as a result of mutations in *CLCNKB* which encode the chloride channel ClC-Kb. The syndrome often has a milder phenotype with later-onset of symptoms, although presentations within the neonatal period have been reported [32]. Patients with Bartter III often have a more severe hypokalaemic alkalosis which is the most likely clinical clue to their underlying genotype [32]. Hypercalciuria and nephrocalcinosis are less frequently associated with the Bartter III phenotype compared to Bartter I and II [32].

Mutations in *BSND*, which encode Barttin (a chaperone protein for ClC-Ka and ClC-Kb) account for Bartter syndrome IV. Sensorineural deafness is part of the Bartter IV phenotype [36]. An association between Bartter syndrome IV and renal impairment has been reported. Patients frequently develop CKD at a very young age, and may not respond to indomethacin, in contrast to those with other aBS genotypes [36].

Although the term Bartter syndrome V is sometimes used in the literature to describe gain-of-function mutations in *CASR* encoding the calcium-sensing receptor (CaSR), according to OMIM classification Bartter syndrome V instead refers to mutations in the *MAGED2* gene, causing an X-linked recessively inherited transient antenatal BS [37]. To avoid this nomenclature issue, CaSR mutations are described separately in the section below.

### 2.6. CASR Mutations

The calcium-sensing receptor (CaSR), expressed in the parathyroid gland and kidney, responds to changes in serum calcium levels, acting to inhibit parathyroid hormone (PTH) secretion and renal tubular calcium reabsorption [38]. A total of 112 mutations of the *CASR* gene have been reported, of which 48 are gain-of-function mutations, including those associated with autosomal dominant hypocalcaemia (ADH)/autosomal dominant hypocalcemic hypercalciuria (ADHH) [38].

Patients with ADH exhibit a biochemical phenotype which includes serum hypocalcemia, low-normal levels of PTH, hypercalciuria, and polyuria [39]. It is essential to distinguish these patients from those with hypoparathyroidism: ADH patients given vitamin D supplementation are likely to develop worsening hypercalciuria, nephrocalcinosis and renal impairment as 1,25-dihydroxyvitamin D upregulates transcription of the *CASR* gene [40]. Instead, a low-normal serum PTH (in contrast to a very low or undetectable level) and low urinary calcium levels in a hypocalcemic patient should prompt clinicians to screen for *CASR* mutations [40]. Patients with ADH should receive treatment for their hypocalcemia only if it is symptomatic, and should aim for symptom control rather than normalised serum calcium levels [40]. Gain-of-function *CASR* mutations that inhibit activity of NKCC2 and ROMK channels in the thick ascending limb (TAL) of loop of Henle have been reported to produce a Bartter-like phenotype, which may include hypokalaemia and hyperreninemic hyperaldosteronism, in addition to hypocalcemia [41,42].

### 2.7. ADCY10 Mutations

Mutations in *ADCY10*, which encodes the soluble adenylyl cyclase gene, have been linked to familial idiopathic hypercalciuria, also known as Absorptive Hypercalciuria, 2 (HCA2) [43]. HCA2 displays autosomal dominant inheritance, with a phenotype including frequent calcium nephrolithiasis [44]. The underlying aetiology of the condition has not been fully delineated, but it is known increased intestinal calcium absorption is responsible for the hypercalciuria seen in these patients. Alongside the common clinical finding of calcium nephrolithiasis, increased renal calcium handling predisposes these patients to nephrocalcinosis formation [44].

### 2.8. Primary Distal Renal Tubular Acidosis

In distal renal tubular acidosis (dRTA), affected patients have a hyperchloremic normal anion-gap metabolic acidosis and alkaline urine (pH > 5.3) [10]. This characteristic biochemical profile arises as a consequence of type A intercalated cells in the collecting duct failing to acidify the urine [45].

Under normal physiological conditions, H^+^ and HCO_3_^−^ are produced within the type A intercalated cells. Mutations in either the vacuolar H^+^-ATPase pump, which excretes H^+^ into the urine, or the chloride bicarbonate counter transporter anion exchanger (AE1), which reabsorbs HCO_3_^−^ into the circulation, account for 85% of known causes of primary dRTA [10]. Mutations in the B1 and A4 subunit of the vacuolar H^+^-ATPase pump, encoded by *ATP6V1B1* and *ATP6V0A4* respectively, demonstrate autosomal recessive transmission and produce a phenotype frequently associated with sensorineural hearing loss as well as the commonly recognised biochemical abnormalities [45]. Mutations in *SLC4A1*, which encodes AE1, can occur with either autosomal dominant or autosomal recessive transmission: autosomal recessive cases are associated with earlier age of symptom onset and a more severe phenotype [10]. Red blood cell abnormalities may also form part of the phenotype for patients with *SLC4A1* mutations [45]. More recently, mutations in Forkhead box protein Il (encoded by *FOXI1)* and WD repeat-containing protein 72 (encoded by *WDR72)* have been recognised as alternative underlying genetic mutations in a small number of families with autosomal recessive inheritance of dRTA [46,47]. At present the underlying genetic defect remains unknown in approximately 15% of cases of primary dRTA [10].

Nephrocalcinosis is an extremely common feature within the phenotype of primary dRTA patients [48]. Calcium phosphate precipitates at higher pH; the alkaline urine of dRTA patients acts as a predisposing factor for nephrolithiasis and nephrocalcinosis formation [45]. In addition, chronic metabolic acidosis leads to excessive bone demineralisation often resulting in hypercalciuria: this also increases the likelihood of nephrocalcinosis developing if patients do not receive early diagnosis and treatment [48].

Treatment strategies for dRTA focus primarily on correcting the underlying metabolic acidosis; patients are maintained on oral potassium citrate which must be taken at regular intervals given its short half-life. This treatment may soon become less cumbersome for patients, as a controlled-release preparation of potassium bicarbonate and potassium citrate designed to be taken twice daily is undergoing phase three trials [10]. Primary distal renal tubular acidosis carries a good prognosis providing prompt diagnosis and treatment initiation are achieved. In patients on treatment with a corrected metabolic acidosis, it has been noted nephrocalcinosis does not progress and their renal function usually remains preserved [10]. However, potassium citrate treatment cannot reverse nephrocalcinosis if already present. Long-term follow-up of dRTA patients should include annual ultrasound screening to monitor for nephrocalcinosis and nephrolithiasis, as well as monitoring of their renal function [45].

### 2.9. Primary Hyperoxaluria

Primary hyperoxaluria describes a group of inborn errors of metabolism where defective liver enzymes involved in glyoxylate metabolism result in excess production of oxalate [49]. The predominant route of oxalate excretion is via the kidneys, with enteric excretion also playing a role when renal function is impaired [50]. In primary hyperoxaluria, excessive oxalate levels supersaturate renal excretion mechanisms, leading initially to calcium oxalate deposition within the kidney, and subsequently systemic oxalosis (deposition of oxalate within other systems) affecting the skeleton, heart, liver and other organs [49].

Currently there are three known types of primary hyperoxaluria which all display autosomal recessive inheritance. Reports of phenotypically similar cases with no proven monogenic cause increase the likelihood of further pathogenic mutations being discovered over time [49]. Recurrent calcium-oxalate nephrolithiasis and nephrocalcinosis are a central feature of primary hyperoxaluria phenotypes. Treatment options and renal prognosis vary considerably between the three genotypes however, making early genetic diagnosis imperative.

Primary hyperoxaluria type 1 (PH1) is the commonest form of primary hyperoxaluria and displays the most severe phenotype, with 50% of patients developing ESRD before the age of 25 [51]. Mutations in *AGXT*, which encode the hepatic alanine-glyoxylate aminotransferase (AGT) enzyme, result in an inability to break down glyoxylate into glycine. Instead, glyoxylate is converted to oxalate, leading to pathological hyperoxaluria [49]. Pyridoxine (vitamin B6) is a co-factor for AGT, and therefore offers a therapeutic target unique to PH1. An estimated 10–40% of PH1 patients respond to pyridoxine supplementation, which can delay progression to ESRF and allow consideration of isolated renal transplantation rather than simultaneous liver/kidney transplantation providing the patient is fully pyridoxine sensitive [51,52]. Early initiation of conservative measures such as high fluid intake and use of potassium citrate may reduce the incidence of nephrolithiasis and delay progression to ESRD [51]. However, due to the risk of systemic oxalosis, renal replacement therapy (RRT) must be initiated once plasma oxalate levels > 30–45 μmol/L, often resulting in patients commencing dialysis at much higher GFR levels than for other renal conditions [52].

PH1 is far more prevalent in developing countries (likely related to higher consanguinity rates) meaning access to liver transplantation is limited [51]. Even in countries where transplantation is more accessible, it carries significant risks for the patient [52]. The evolution of our treatment of this condition therefore hinges upon identifying therapeutic targets which do not necessitate organ transplantation to replace the defective enzyme. Trials utilising the oxalate-metabolising bacterium *Oxabacter formigenes* to increase gut excretion of oxalate and reduce urinary oxalate excretion reached phase II/III trials but did not significantly reduce urinary oxalate excretion [50]. Animal studies have shown oxalate decarboxylase enzymes can effectively reduce urinary oxalate levels, providing an alternative potential future therapy for reduction of calcium-oxalate nephrocalcinosis in primary hyperoxaluria [53,54].

Primary hyperoxaluria type 2 (PH2) occurs as a result of mutations in *GRHPR* which encodes the glyoxylate reductase/hydroxypyruvate reductase enzyme. PH2 exhibits a less severe phenotype, with lower incidence of nephrolithiasis and nephrocalcinosis. It is rare for PH2 patients to progress to ESRD or develop systemic oxalosis [51].

Primary hyperoxaluria type 3 (PH3) account for only 10% of PH patients [49]. Mutations in *HOGA1*, which encodes the hepatic enzyme 4-hydroxy-2-oxoglutarate aldolase, underlie the condition. PH3 has a milder phenotype, with lower nephrolithiasis burden, and nephrocalcinosis and renal impairment being less common

### 2.10. GDNF Mutations

Medullary sponge kidney (MSK) is a congenital disorder resulting in ectatic collecting ducts within one or both kidneys. The clinical sequelae of nephrolithiasis, nephrocalcinosis, recurrent urinary tract infections and urinary acidification defects are often not apparent until adulthood [55]. The underlying pathogenesis of MSK is yet to be fully elicited. Whilst once considered a sporadic disorder, evidence of MSK cases showing likely autosomal dominant inheritance have emerged, intensifying the search for underlying genetic causes [56]. Identification of mutations in *GDNF*, which encodes glial cell-derived neurotrophic factor, have now been identified in some MSK patients and may account for the underlying pathophysiological process in a subset of MSK patients [57].

## 3. Other Genetic Conditions That May Feature Nephrocalcinosis within Their Clinical Phenotype

Hypercalciuria acts as a predisposing factor for medullary nephrocalcinosis formation; any condition associated with excess urinary calcium excretion may lead to cases of nephrocalcinosis. Hypercalciuria and nephrocalcinosis have been described in association with the inherited conditions Wilson’s disease, Williams-Beuren syndrome and cystic fibrosis.

Wilson’s disease, a rare autosomal recessive condition characterised by *ATP7B* mutations leading to defective copper excretion, can feature a renal Fanconi syndrome [58,59]. In Wilson’s disease patients exhibiting hypercalciuria, nephrolithiasis and nephrocalcinosis have been described [59].

Williams-Beuren syndrome is a developmental disorder occurring as a result of a microdeletion on the q arm of chromosome 7. The classical phenotype includes a distinctive facial appearance, intellectual disability and cardiovascular problems. However, patients with this syndrome have been found to be at increased risk of hypercalcaemia; the combination of resulting dehydration and hypercalciuria have led to cases of nephrocalcinosis being described [60].

Cystic fibrosis, caused by mutations in the *CFTR* gene, has been associated with hypercalciuria and microscopic nephrocalcinosis. However, the patients described did not demonstrate any signs of renal dysfunction associated with their microscopic nephrocalcinosis [61].

Amelogenesis Imperfecta describes a group of inherited enamel defects; there have been case reports of nephrocalcinosis detection in some of these patients in association with a distal renal tubular acidosis. However, in contrast to Wilson’s disease, Williams-Beuren syndrome and cystic fibrosis, these patients have not demonstrated hypercalciuria, indicating different underlying pathophysiology [62].

## 4. Conclusions

Monogenic causes of nephrocalcinosis account for a rare set of conditions with a low population frequency. However, the clinical course for affected patients is very different depending on their underlying genotype. Accurate, prompt diagnosis allows early initiation of conservative measures, which in some cases can halt nephrocalcinosis or delay progression of renal impairment. Establishing the underlying genetic mutation also allows accurate prognostic information to be given and can help facilitate screening of other family members. As our understanding of these rare inherited conditions increases, it is hoped further treatment targets that address underlying enzymatic/protein defects will emerge. However, at present for the majority of cases treatment strategies focus upon supportive treatments to correct biochemical parameters, and careful monitoring of disease progression.

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
