# Peer review of "Nephrocalcinosis: A Review of Monogenic Causes and Insights They Provide into This Heterogeneous Condition"

_ijms, 2020, doi:10.3390/ijms21010369_

Round 1

Reviewer 1 Report

Page 2 line 69 : "The morbidity........focus predominantly on slowing progression of associated CKD" This statement is not entirely correct. There are specific treatment options for different underlying diseases eg changes in diet or medications such as potassium citrate, thiazides. Please modify

Page 3 line 120: "their role is often limited by unacceptable side effects" the word often is to strong. I would prefer "in some patients"

page 6 line 236 "prior to classic hypokalemia developing" prior to development of classic hypokalemia?

page 8 line 294 should say "...leads to excessive bone demineralizaton"

Table: in general: please stick to the OMIM nomenclature

The gene affected causing Dent disease 2 is OCRL; please mention the name Lowe syndrome

SLC34A1: please mention that the type of IIH which is caused by mutations in this gene is called infantile hypercalcemia type 2; HCINF2 (hypercalemia, infantile, 2)

Likewise CYP24A1 mutation causes HCINF1...

Please also include Wilson's disease as a monogenic cause of nephrocalcinosis (Hoppe B Nephron 1993)

Please include Wiliams Beuren syndrome even though it is not strictly monogenic

Please mention that there is a problem with the nomenclature of Bartter syndrome type 5. Most people would say that mutations in MAGED5 cause BARTS5

Reviewer 2 Report

This is a very interesting review paper grouping the monogenic causes of nephrocalcinosis.

I have some minor comment, which may help ameliorate the quality of the manuscript:

In the first paragraph in the Introduction section, it should be added that medullary nephrocalcinosis may be subdivided according to the degree of echogenicity as medullary NC grade I-III (Dick PT, Shuckett BM, Tang B, Daneman A, Kooh SW. Observer reliability in grading nephrocalcinosis on ultrasound examinations in children, Pediatr Radiol. 1999 Jan;29(1):68-72) In the paragraph of Dent disease, it should be mentioned that other disease presented as Fanconi syndrome may also involve nephrocalcinosis, such as Lowe syndrome, caused by variants in the OCRL gene (Bökenkamp A, Ludwig M. The oculocerebrorenal syndrome of Lowe: an update. Pediatr Nephrol. 2016 Dec;31(12):2201-2212) and cystinosis, caused by mutations in the CTNS gene (Elmonem MA, Veys KR, Soliman NA, van Dyck M, van den Heuvel LP, Levtchenko E. Cystinosis: a review. Orphanet J Rare Dis. 2016 Apr 22;11:47) Other monogenic causes of nephrocalcinosis that need to be included are: Autosomal dominant hypocalcemic hypercalciuria (ADHH), caused by mutations in the calcium-sensing receptor (CaSR) gene Cystinuria Familiar idiopathic hyperalciuria, caused by mutations in ADCY10 gene Medullary sponge kidney (MSK) causes by mutations in GDNF gene Less common genetic diseases with possible nephrocalcinosis are:Cystic fibrosis, X-linked hypophosphatemic rickets (PHEX gene), Williams Beuren syndrome, Wilson syndrome, MacGibbon-Lubinsky syndrome and Liddle syndrome. An algorithm based on the serum calcium, 25 and 1,25 (OH)D, phosphorus, magnesium, PTH, blood gas result, urine anion gap, urine amino-acids and result of extra-renal manifestations may be illustrated to help the clinician in the differential diagnosis of nephrocalcinosis.
